# Pentosan polysulfate regulates hepcidin 1-facilitated formation and function of osteoclast derived from canine bone marrow

**Suranji Wijekoon**[ID][☯]\*, **Takafumi Sunaga**[‡], **Yanlin Wang**[‡], **Carol Mwale**[‡], **Sangho Kim**[‡], **Masahiro Okumura**[☯]

Laboratory of Veterinary Surgery, Department of Veterinary Clinical Sciences, Graduate School of Veterinary Medicine, Sapporo, Hokkaido, Japan

☯ These authors contributed equally to this work.
‡ TS, YW, CM and SK also contributed equally to this work.
\* suranjisk@gmail.com

**Data Availability Statement:** Data Availability: All relevant data are within the paper and its Supporting information files.

## Abstract

Hepcidin which is the crucial regulator of iron homeostasis, produced in the liver in response to anemia, hypoxia, or inflammation. Recent studies have suggested that hepcidin and iron metabolism are involved in osteoporosis by inhibiting osteoblast function and promoting osteoclastogenesis. Pentosan polysulfate (PPS) is a heparin analogue and promising novel therapeutic for osteoarthritis (OA). This study was undertaken to determine whether PPS inhibits hepcidin-facilitated osteoclast (OC) differentiation and iron overload. Canine (n = 3) bone marrow mononuclear cells were differentiated to OC by macrophage colony-stimulating factor and receptor-activator of nuclear factor kappaB ligand with the treatment of hepcidin1 (200, 400, 800, 1200 nmol/L) and PPS (1, 5, 10, 20, 40 μg/mL). Differentiation and function of OC were accessed using tartrate-resistant acid phosphate staining and bone resorption assay while monitoring ferroportin1 (FPN1) and iron concentration by immunocytochemistry. Gene expression of OC for cathepsin K (CTK), matrix metallopeptidase-9, nuclear factor of activated-T-cells cytoplasmic 1 and FPN1 was examined. Hepcidin1 showed significant enhancement of OC number at 800 nmol/L (p<0.01). PPS impeded hepcidin-facilitated OC at 1, 5 and 10 μg/mL and reduction of resorption pits at 5 and 10 μg/mL (p< 0.01). All OC specific genes were downregulated with PPS, specifically in significant manner with CTK at higher concentrations. However, heparin induced FPN1 internalization and degradation was inhibited at higher concentrations of PPS while restoring iron-releasing capability of OC. We demonstrate for the first time that PPS is a novel-inhibitor of hepcidin-facilitated OC formation/function which might be beneficial for treatment of OA and osteoporosis.

**Funding:** The research was funded by Japan Racing Association. The funders had no role in study design, data collection and analysis, decision to publish, or preparation of the manuscript.

**Competing interests:** The authors have declared that no competing interests exist.

## Introduction

Hepcidin is a recently discovered cysteine-rich cationic, small endogenous peptide hormone synthesized in hepatocytes and involved in the regulation of iron homeostasis [1, 2]. Secreted hepcidin inhibits iron transport by binding to the iron export channel ferroportin which is located in the basolateral plasma membrane of gut enterocytes and the plasma membrane of reticuloendothelial cells (macrophages), while breaking down the ferroportin in to lysosomes [3]. Further, hepcidin inhibits the release of iron into the circulation by regulating its associated receptor ferroportin 1 (FPN 1) [1].

Hepcidin expression is exaggerated in chronic inflammatory conditions due to cancer, infectious autoimmune disorders, hypoxia, and anemia [4–9]. Interleukin 6 (IL-6) plays a key role in inflammatory induction of hepcidin and ultimately causes hypoferremia due to ferroportin degradation and iron sequestration in tissue macrophages [10]. The IL-6 blockers have been shown as highly effective on the anemia of chronic disease, commonly observed in rheumatoid arthritis (RA). Several studies have focused on the role of hepcidin in RA and found that the circulating serum hepcidin level is reported to increase in the RA patients with anemia of chronic disease (ACD) [11–13]. With understanding of all those factors hepcidin is recognized as a major inducer of ACD in patients with RA.

Some recent reports have indicated that iron metabolism can affect bone metabolism. Elevated ferritin level/iron overload is a risk factor for progressive bone loss in healthy postmenopausal women and middle-aged men [14]. However, the direct effects of hepcidin on bone metabolism are still unknown although it shows potential of pharmacological target. Bone metabolism in the body involves osteoblastic bone formation as well as osteoclastic bone resorption. Previous studies showed that hepcidin could increase intracellular iron and calcium levels and promote mineralization in osteoblasts [15–17] and osteoclasts (OC) [1].

Pentosan polysulfate (PPS) is a semi-synthetic sulfated polysaccharide drug manufactured from European beech wood hemicellulose by sulfate esterification. Average molecular weight of PPS is around 5700 Da [18]. From the results of previous *in vitro* and *in vivo* studies, the spectrum of pharmacological activities exhibited by PPS would qualify it as disease-modifying osteoarthritis drugs [19] due to its ability of preserving the integrity of the articular cartilage and bone whilst improving the characteristic of the synovial joint fluid [20–25]. Further we have previously identified that the inhibitor effect of PPS on formation and function of bone marrow derived-OC which was differentiated with the presence of receptor activator of nuclear factor kappa B ligand (RANKL) and macrophage colony-stimulating factor (M-CSF) [26].

Bone homeostasis between bone resorption by OC and bone formation by chondrocytes/osteoblasts is tightly orchestrated to preserve skeletal health and integrity throughout life [27]. Osteoclasts originate from the monocyte/macrophage hematopoietic lineage, whereas osteoblasts originate from multipotent mesenchymal stem cells. Osteoclastogenesis is a tightly regulated process by diverse cytokines, steroids, and lipids [28, 29]. Among them, M-CSF and RANKL predominantly involve differentiation, maturation, and function of this giant cell [30, 31]. However, excessive activation of the immune mediators at the inflammatory conditions can enhance the OC production and eventually the severe bone erosion can be occurred [32]. Therapeutic interventions targeting osteoclastogenesis might enable it to restore bone mass in arthritic patients.

This study was undergone to find out whether PPS is able to control the formation and function of hepcidin 1 treated OC and thereby intracellular iron concentration. To the best of our knowledge, the present study is the first attempt to identify the effect of PPS on hepcidin facilitated differentiation and intracellular iron concentration in bone marrow derived OC.

We hypothesized that the PPS, which carry different effects by improving the symptoms of osteoarthritis and RA, would be more likely to have an inhibitory effect on hepcidin.

## Material and methods

### Preparation of sample collection site

Proximal femur of healthy beagle dogs (n = 3) was used to collect the 5 mL of bone marrow samples in to 10 mL syringe containing 1 mL Dulbecco's modified eagle's medium (DMEM, Life technologies, New York, USA) and 1000 U/mL of heparin (Nipro, Osaka, Japan). The use of all samples from healthy experimental Beagle dogs (mean age: 12.9 months; range: 12–14 months) was in accordance with Hokkaido University Institutional Animal Care and Use Committee guidelines (approval number: 21–0022). Briefly, dogs were put under general anesthesia induced with propofol (Intervet, Tokyo, Japan) at 6 mg/kg intravenously and maintained on isoflurane (Intervet) and oxygen. Meloxicam (Boehringer-Ingelheim Animal Health, Tokyo, Japan) at 0.2 mg/kg subcutaneously was administered for pain management. The bone marrow aspiration site was aseptically prepared by clipping the hair around the proposed site of collection and scrubbed with 70% ethanol and then povidone iodine.

### Osteoclastic differentiation from canine bone marrow

Separation of bone marrow mononuclear cell (BMMs) fraction was done and preceded as described previously [33, 34]. Briefly, BMMs were obtained by density gradient centrifugation over lymphoprep (Axis-sheild PoC AS, Oslo, Norway) to remove red blood cells. Isolated BMMs cell fraction ($5 \times 10^6$ cells/mL) was incubated with DMEM containing penicillin/streptomycin (100 units/mL, Wako pure chemical, Tokyo, Japan) and 10% heat-inactivated fetal bovine serum (FBS, Nichirei Bioscience INC., Tokyo, Japan) for 24 h to separate the non-adherent from adherent cells. Non-adherent cells were collected as a source of immature OC precursors, suspended in DMEM, counted, seeded on 48-wells plates (Corning, New York, USA) at $2 \times 10^5$ cells/well, and cultured in DMEM with the presence of 20 ng/ml recombinant human M-CSF (Invitrogen, Maryland, USA) for 3 days. After 3 days, adherent cells were used as OC precursors after washing out the non-adherent cells, including lymphocytes and further cultured in the presence of 25 ng/mL M-CSF, 50 ng/mL recombinant human RANKL (Sigma-Aldrich, St Louis, Missouri, USA) and 0, 200, 400, 800 nmol/L hepcidin 1 to generate osteoclast-like multinucleated giant cells. The selected concentrations of hepcidin are within the previously proved non-cytotoxic range for mouse macrophages [1]. The cells were treated with 1, 5, 10, 20 μg/mL concentration of PPS (Cartrophen Vet-Biopharm-100 mg/ ml, New South Wales, Australia) for 1-week. The selected concentrations of PPS are within the previously proved non-cytotoxic range for bone marrow derived cells [35]. Triplicate cultures for each concentration of PPS were maintained by changing the media in every 48 h ensuring their constancy of concentrations.

### Tartrate-resistant acid phosphate (TRAP) staining

Cultured BMMs with M-CSF and RANKL in the presence of 0, 200, 400, 800 nmol/L hepcidin and 1, 5, 10, 20 μg/mL PPS were subjected to TRAP stain (Cosmo Bio Co., LTD, Tokyo, Japan) after 7 days. Cells were washed with 1% phosphate buffered saline (PBS) and fixed with 10% formalin neutral buffer solution for 5 min at room temperature. After washing with 500 μL deionized water 3 times, cells were stained for TRAP according to the manufacturer's instructions. Cells containing ≥3 nuclei were considered as OC and counted.

## Resorption pit formation assay

Non-adherent cells, collected from BMMs fraction of 3 dogs were cultured at $2 \times 10^5$ cells/well density on calcium phosphate (CaP-coated) bone resorption assay plate 48 (PG Research, Tokyo, Japan). The cells were maintained in DMEM with the presence of 20 ng/ml recombinant human M-CSF (Invitrogen, Maryland, USA) for 3 days in triplicate cultures. After 3 days, adherent cells were used as OC precursors after washing out the non-adherent cells and further cultured in the presence of 25 ng/mL M-CSF, 50 ng/mL recombinant human RANKL, hepcidin (0, 200, 400, 800 nmol/L) and PPS (5 μg/mL). After 7 days, the CaP-coated plate was treated with 5% sodium hypochlorite (Sigma-Aldrich, St Louis, Missouri, USA) for 5 min according to the manufacturer's instructions. The resorption pit area was analyzed and counted by Image-J software (Image J software version 1.43, National Institute of Health).

## Preparation of cell extracts and analysis

Total RNA and protein were extracted using TRIZol reagent (Invitrogen, Life Technologies, Carlsbad, CA, USA) according to the manufacturer's protocol. Total RNA was quantified by spectrophotometry at 260 nm using Biowave DNA-WPA, 7123 V1.8.0 (Biochrom, Cambridge, UK) and stored at -20 ˚C until use. RNA with a 260/280 nm ratio in the range 1.8–2.0 was considered high quality and total of 500 ng RNA was reverse transcribed (RT) into cDNA using ReverTra Ace qPCR RT Master Mix (Toyobo Co, Osaka, Japan) and amplified by PCR using TaKaRa Ex taq (TaKaRa Bio, Tokyo, Japan) according to manufacturer's recommended protocol. This technique was employed to amplify mRNAs specific for cathepsin k (CTK), matrix metallopeptidase-9 (MMP9), nuclear factor of activated T-cells cytoplasmic 1 (NFATc1) and ferroportin 1 (FPN1). The PCR conditions were an initial denaturation of 94 ˚C for 1 min followed by 35 cycles of 94 ˚C for 30 s, 58 ˚C for 30 s and 72 ˚C for 30 s and then a finishing step of 72 ˚C for 1 min. Gel electrophoresis was performed to detect mRNA bands. Quantitative real-time PCR (qPCR) was performed with KAPA SYBR FAST qPCR kit (KAPA biosystems, Woburn, MA, USA) to determine the relative mRNA expression by the two step method. The qPCR conditions were an initial denaturation of 95 ˚C for 20 s followed by 40 cycles of 95 ˚C for 3 s and 60 ˚C for 20 s then a pre-melt condition of 60 ˚C for 90 s followed by a final melt step. The standard curve method was used to determine the relative mRNA quantification. All PCR reactions were validated by the presence of a single peak in the melt curve analysis and single band on gel electrophoresis. The amount of 2 μL of cDNA template was added to each 10 μL of premixture with specific primers. The following primer sets were used: cathepsin k, `5'- ACCCATATGTGGGACAGGAT-3'` (forward) and `5'-TGGAAAGAGGTCAGGCTTGC-3'` (reverse); MMP9, `5'-GGCAAATTCCAGACCTTTGA-3'` (forward) and `5'-TACACGCGAGTGAAGGTGAG-3'` (reverse); NFATc1, `5'-CACAGGCAAGACTGTCTCCA-3'` (forward) and `5'-TCCTCCCAATGTCTGTCTCC-3'` (reverse); FPN1, `5'-CAGTCTATGGGCTGGTGGTG-3'` (forward) and `5'-TCTGGATCGTGATGGCAGTG-3'` (reverse); GAPGH, `5'-CTGA ACGGGAAGCTCACTGG-3'` (forward) and `5'-CGATG CCTGCTTCACTACCT-3'` (reverse). All reactions were normalized to the housekeeping gene b-Actin.glyceraldehyde-3-phosphate dehydrogenase (GAPDH).

## Immunocytochemical detection of ferroportin 1 (FPN 1)

Osteoclast precursors resulting from canine bone marrow cells ($2 \times 10^5$ cells) were cultured in 8-wells culture slide (Iwaki, Tokyo, Japan) in 200 μL of DMEM, 10% FBS with OC differentiation factors (25 ng/mL M-CSF, 50 ng/mL recombinant human RANKL). Cells were treated separately and combined with hepcidin (0, 200, 400, 800, 1200 nmol/L) and PPS (5, 10, 20, 40 μg/mL) for 20 hours. The cells were fixed with 4% paraformaldehyde for 20 min. After

being blocked in 1% bovine serum albumin (BSA) for 45 minutes at room temperature, the cells were then incubated with primary antibodies (rabbit anti-ferroportin 1, 1:50, Lifespan Biosciences, Seattle, Washington, USA) in a humid chamber at 4 ˚C overnight. After being washed three times with PBS, the cells were incubated with a secondary antibody (goat anti-rabbit, 1:500, Alexa Fluor 488, Abcam, Cambridge, UK) away from light at room temperature for 45 minutes. After washing three times with PBS, the cells were stained with DAPI. The OC were observed using laser scanning confocal microscope (Zeiss, Illinois, USA).

### Immunofluorescent analysis of intracellular iron concentration

Fluorescence of the Phen Green FL is quenched upon binding iron ion, so the change of emission intensity of the indicator is correlated with iron concentration. The weaker the fluorescence intensity is an indicator of the higher the concentration of intracellular iron [1]. Osteoclast precursors cells were seeded in an 8-wells culture slide of $2 \times 10^5$ cells/well. After cells adhered to the coverslips, the medium was replaced with fresh medium containing 25 ng/mL M-CSF, 50 ng/mL RANKL and 800, nmol/L hepcidin 1 and 20 μg/mL PPS and allowed to incubate for 20 h. After treatment, the cells were washed twice with PBS and incubated with Phen Green FL (Carlsbad, California, USA) away from light at 34˚C in a humidified atmosphere containing 5% $CO_2$ for 30min. Unbound fluorescent indicator was removed by washing with PBS two times. A confocal laser scanning microscope was used to record the signal intensity from Phen Green FL, with excitation at 488 and emission at 521 nm.

### Data analysis

Data were analyzed using Statistical Package for the Social Sciences v.16 (SPSS inc., Chicago, IL, USA). Statistical significance of quantitative qPCR data, number of OC and resorption pits were determined by analysis of variance (ANOVA), comparing the mean values of the treatments. Where significant differences were observed, multiple comparison of group means was performed using Post Hoc Bonferroni. The results were considered significant at a 95% confidence level ($p < .05$). All quantitative results are presented as mean ± SE.

## Results

### Hepcidin 1 facilitate M-CSF and RANKL-induced osteoclastogenesis

Effect of hepcidin 1 on M-SCF and RANKL treated BMMs were evaluated. Dose-dependent stimulatory effect of hepcidin 1 was observed up to the 800 nmol/L concentration by counting the TRAP positive multinucleated cells (≥3 nuclei) seeded in 48 wells plates (Fig 1A). However, TRAP-stained cell number reduced at the 1200 nmol/L concentration. Significant enhancement of OC number was detected at the 800 nmol/L concentration (p<0.05) (Fig 1B).

### PPS inhibits osteoclastogenesis

Effects of different concentrations of PPS were evaluated over the BMMs treated M-CSF and RANKL. Dose dependent inhibition of TRAP-stained multinucleated cells (≥3 nuclei) (Fig 2A) were detected significantly at all the concentrations of PPS from 1 (p<0.001), 5, 10, 20 μg/mL (p<0.005) (Fig 2B).

### PPS inhibits hepcidin-facilitated OC formation and function in dose dependent manner

The effect of different concentrations of PPS (1, 5, and 10 μg/mL) on OC differentiation from BMMs stimulated with M-CSF, RANKL and hepcidin 800 nmol/L was evaluated. The number

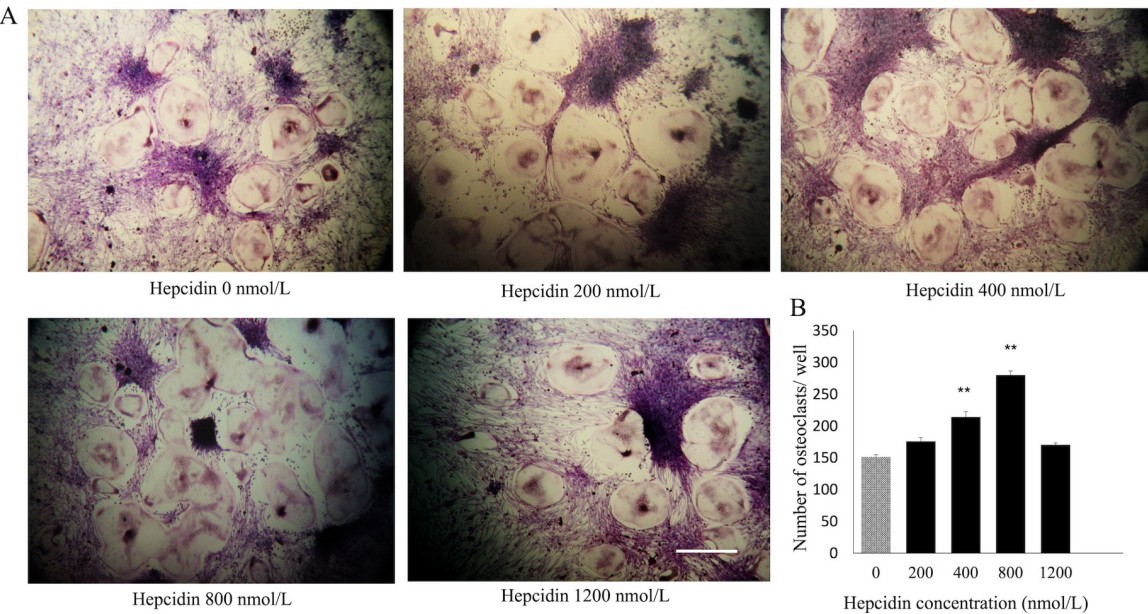

**Fig 1. Shows effect of hepcidin 1 on canine OC differentiation.** (A) The cells were treated with various concentrations of hepcidin followed by M-CSF (20 ng/mL) and RANKL (50 ng/mL) for 7 days. The cells were stained for TRAP stain and TRAP-positive cells ($\geq 3$ nuclei) were counted. Scale bar- 200 μm. (B) Bar graphs show the number of OC cells/well. Data are representative of three independent experiments and expressed as means ± SE. Means with *are significantly different from 0 μg/mL of hepcidin (*$p < 0.05$, **$p < 0.01$).

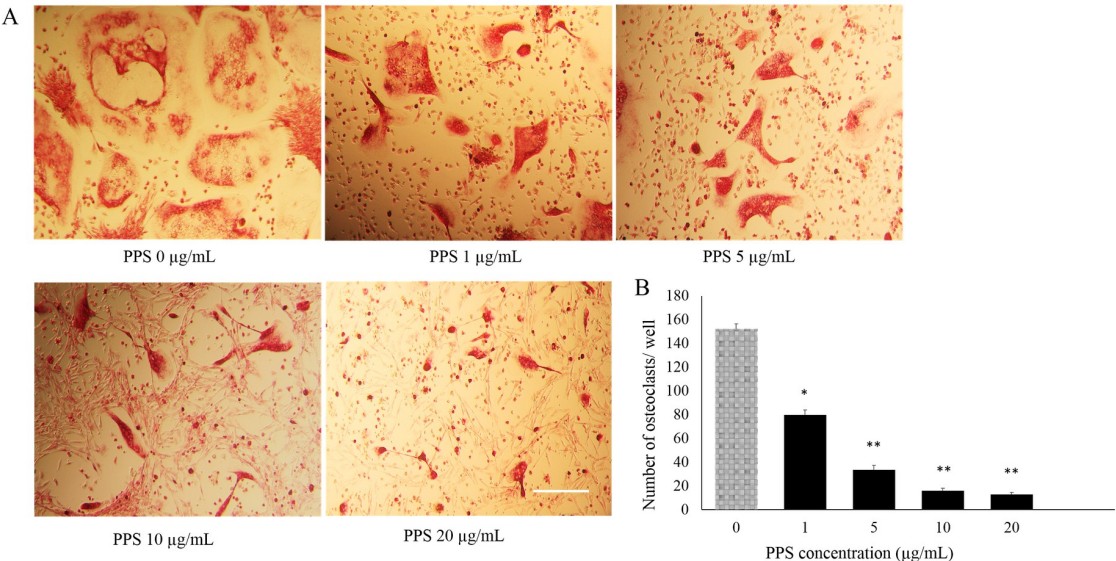

**Fig 2. Shows inhibitory effect of PPS on canine OC differentiation.** (A) The cells were treated with various concentrations of PPS followed by M-CSF (20 ng/mL) and RANKL (50 ng/mL) for 7 days. The cells were stained for TRAP stain and TRAP-positive cells ($\geq 3$ nuclei) were counted. Scale bar- 100 μm. (B) Bar graphs show the number of OC cells/well. Data are representative of three independent experiments and expressed as means ± SE. Means with *are significantly different from 0 μg/mL of PPS (*$p < 0.05$, **$p < 0.01$).

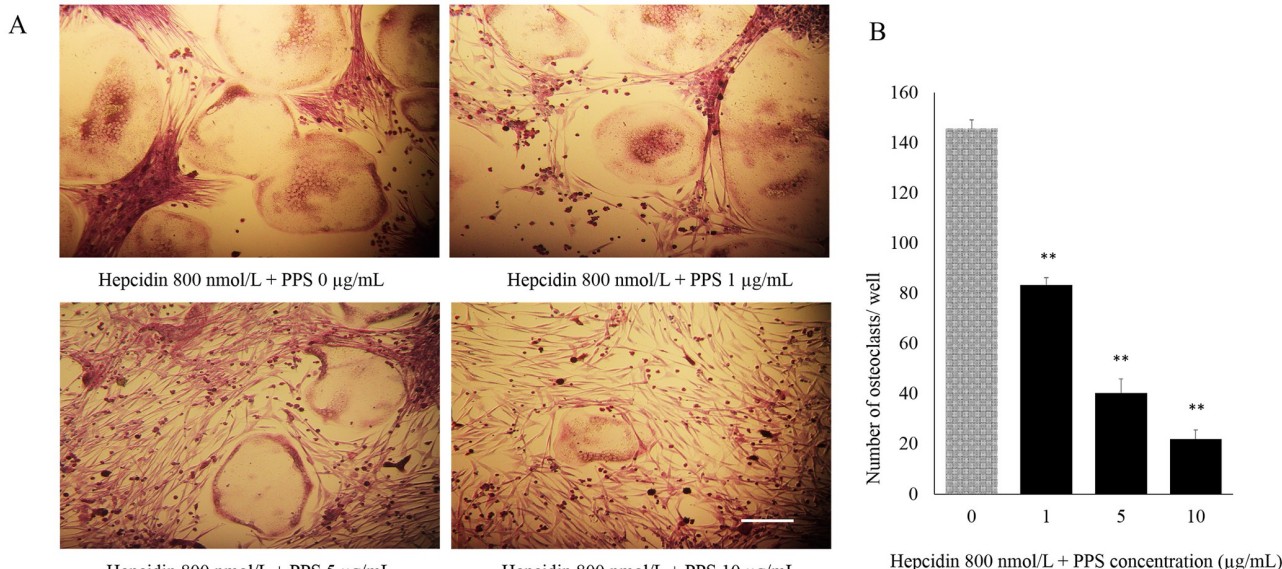

**Fig 3. Inhibitory effect of PPS on OC treated with hepcidin 1.** The cells were treated with various concentrations of PPS followed by 800 nmol/L hepcidin 1, M-CSF (20 ng/mL) and RANKL (50 ng/mL) for 7 days. (A) The cells were stained for TRAP stain and TRAP-positive cells (≥3 nuclei) were counted. Scale bar- 200 μm. (B) Bar graphs show the number of OC cells/well. Data are representative of three independent experiments and expressed as means ± SE. Means with *are significantly different from 0 μg/mL of PPS (*$p < 0.05$, **$p < 0.01$).

of TRAP-positive multinucleated cells (≥3 nuclei) generated in 48 well plate was reduced (Fig 3A) with the administration of PPS at the concentration of 1, 5 and 10 μg/mL in significant manner ($p < 0.05$) compared to the control samples those were not treated with PPS (Fig 3B). The effect of PPS on hepcidin-facilitated OC function was assessed via counting the bone resorption pits formed by OC generated from 3 dogs. Cells were plated on CaP-coated plates and stimulated with M-CSF, RANKL and hepcidin 800 nmol/L in the presence and absence of PPS. Cells stimulated with M-CSF, RANKL and hepcidin formed a number of resorption pits suggesting that the bone resorption activity of RANKL-treated cells made them into functionally active state resembling OC. The concentrations of 5 and 10 μg/mL PPS significantly reduced the formation of resorption pits (Fig 4A) in number and in overall area compared with treatment with M-CSF, RANKL and hepcidin alone (Fig 4B). Gene expression of Cathepsin K, MMP-9, NFATc1 and FPN1 was investigated (Fig 4C). Quantitative qPCR data showed that PPS at 5–10 μg/mL significantly downregulates the gene expression ($p<0.05$) level of CTK while inhibiting expression level of MMP-9 and NFATc1 in concentration-dependent manner (Fig 4D). However, the expression of FPN1 was upregulated toward the higher concentrations of PPS.

## PPS inhibits the hepcidin-induced FPN1 internalization and degradation

Canine bone marrow derived OC were treated with rabbit anti-ferroportin 1 after 20 hours treatment of hepcidin (200, 400, 800 and 1200 nmol/L) to visualize the expression of FPN1 protein. Immunofluorescence showed the expression of FPN1 protein in OC treated MCSF and RANKL (Fig 5A). The intensity of fluorescence was reduced towards the 800 and 1200 nmol/L concentration of hepcidin (Fig 5B). Separate groups of OC were teared with different concentrations of PPS for overnight after treatment of 20 hours of 800 nmol/L hepcidin. However, heparin induced FPN1 internalization and degradation was inhibited with higher

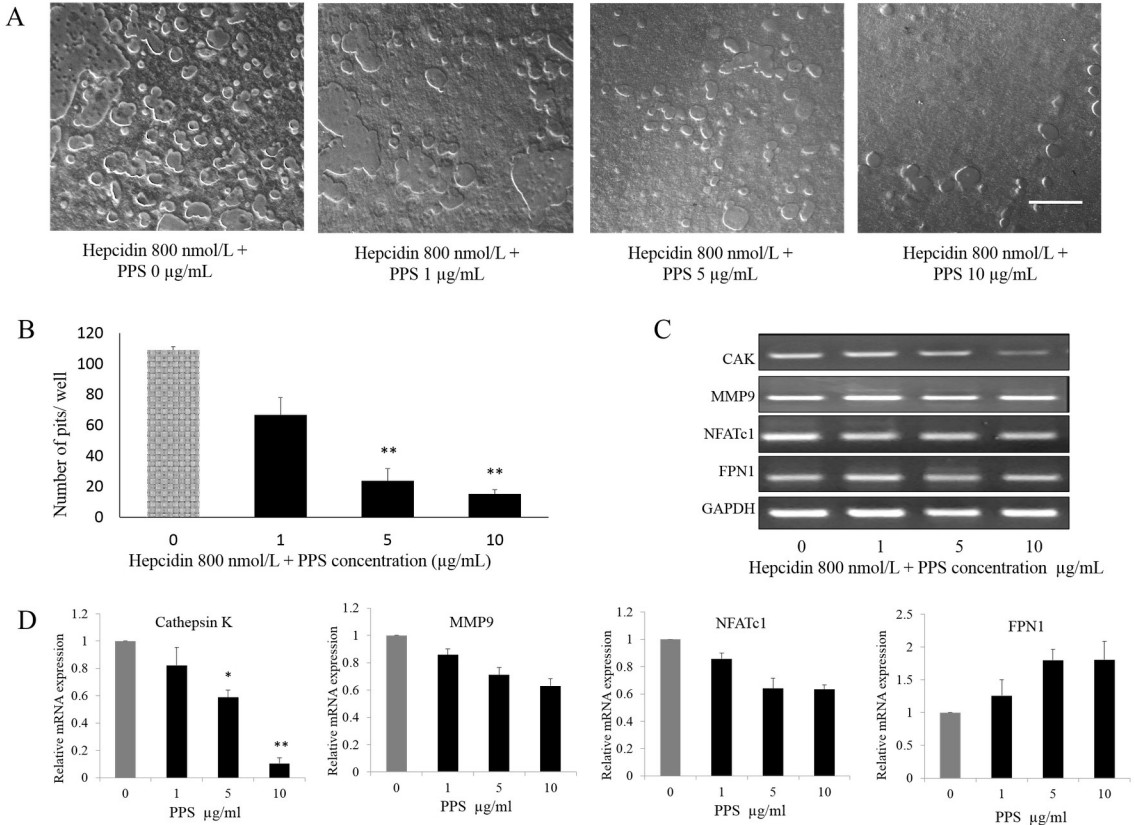

**Fig 4. PPS inhibits bone resorption and OC specific genes.** Canine BMMs, cultured with M-CSF (20 ng/mL), RANKL (50 ng/mL) and 800 nmol/L hepcidin 1 for 7 days with or without indicated doses of PPS. (A) The cells were washed and the resorption pits were counted. (B) The numbers of pits were analyzed with Image-J software. Scale bar- 200 μm. Column indicates means ± SE of three experiments performed in triplicate. (C) and (D) Gene expression of CTK, MMP9, NFATc1 and FPN1 was investigated. Quantitative qPCR showed that PPS downregulates the gene expression of MMP9, NFATc1 and CTK (CTK at 5 and 10 μg/mL of PPS; p<0.05) while upregulating FPN1 expression in concentration manner. Data expressed as mean ± SE for each PPS concentration after normalizing for the expression of the GAPDH. Means with *are significantly different from 0 μg/mL of NaPPS (*p < 0.05, **p < 0.01).

concentrations of PPS (5, 10, 20 and 40 μg/mL). Higher the fluorescence intensity was visualized toward the 40 μg/mL PPS (Fig 5C and 5D).

## PPS repressed the hepcidin-induced intracellular iron accumulation

Fluorescence of the Phen Green FL is quenched upon binding iron and emission intensity was weakened when the iron concentration is high. Fluorescence images showed that intracellular iron concentration was increased (weakened fluorescence intensity) with hepcidin 800 nmol/L compared to the untreated cells (Fig 5E and 5F). However, fluorescence images have proven the inhibitory effect of PPS on hepcidin-induced iron accumulation by visualizing greater intensity of fluorescence (Fig 5G).

## Discussion

Hepcidin-centered therapeutic studies are a fascinating field of research to regulate the iron homeostasis, anemia of inflammation and bone metabolism. Numerous therapeutic modalities which are identified targeting the property of cytokines-based hepcidin repression seem

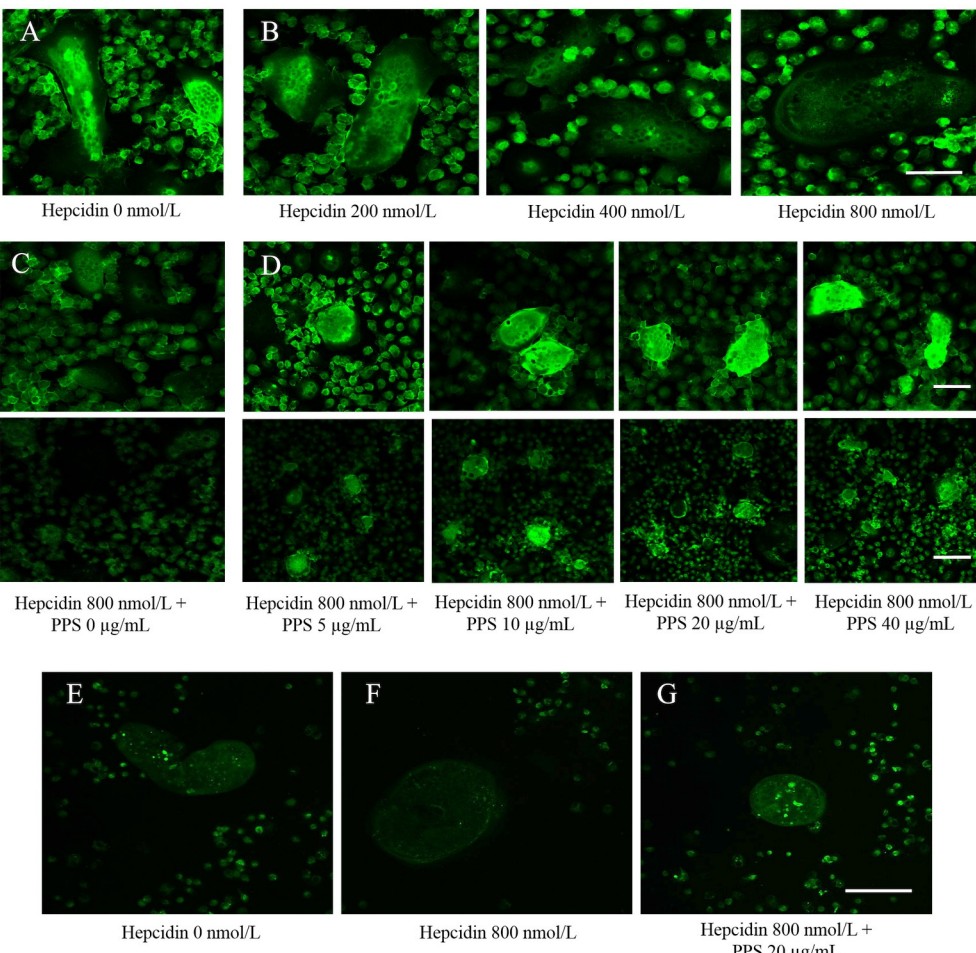

**Fig 5. Immunofluorescence images.** (A) Immunofluorescence analysis of ferroportin 1 (FPN1) protein. Canine bone marrow derived OC were treated with rabbit anti-ferroportin 1 after 20 hours treatment of hepcidin (200, 400 and 800 nmol/L) to visualize the expression of FPN1 protein. The figure shows the strong detection of localization of FPN1 at the membrane of hepcidin untreated OC (control). (B) The green fluorescence intensity significantly weakened toward the higher concentration of hepcidin (800 nmol/L). Scale bar- 50 μm. (C) PPS inhibits the hepcidin-induced FPN1 internalization and degradation. OC were treated with rabbit anti-ferroportin 1 after 20 hs treatment of PPS (5, 10, 20 and 40 μg/mL) and hepcidin 1 (800 nmol/L) to visualize the expression of FPN1 protein. The figure shows the strong inhibition of localization of FPN1 at the membrane of hepcidin treated OC (control). (D) Heparin induced FPN1 internalization and degradation was inhibited with higher concentrations of PPS (5, 10, 20 and 40 μg/mL). Higher the fluorescence intensity was visualized toward the 40 μg/mL PPS. Scale bars- 50 and 100 μm. (E) Confocal microscopy analysis of iron concentration in OC. Cell medium was replaced with fresh medium containing 25 ng/mL M-CSF, 50 ng/mL RANKL and 800, nmol/L hepcidin 1 and 20 μg/mL PPS and allowed to incubate for 20 h. Fluorescence of the Phen Green FL is quenched upon binding iron and emission intensity was weakened when the iron concentration is high. E, shows high fluorescence intensity in untreated cells implying low intracellular iron. (F) Fluorescence images showed that intracellular iron concentration was increased (weakened fluorescence intensity) with hepcidin 800 nmol/L compared to the untreated cells. (G) fluorescence images have proven the inhibitory effect of PPS on hepcidin-induced iron accumulation by visualizing greater intensity of fluorescence at 20 μg/mL.

effective over the patients of OA and RA [36, 37]. The present study demonstrates for the first time that PPS is a novel inhibitor of hepcidin-facilitated OC formation and function from bone marrow-derived stem cells. With the multiple facets of action of PPS, this novel finding would be upgraded the uses of DMOARs and specially, PPS for the betterment of OA and RA associated anemia and iron imbalance.

Similar to the finding of mouse macrophage-generated OC culture [1], our findings demonstrate that hepcidin facilitate MSCF and RANKL induced osteoclastogenesis from canine macrophages in dose dependent manner up to 800 nmol/L concentration. The fact that understands in this study is concentration over the 800 nmol/L was no able to further facilitate canine OC formation which is controversy with previous mouse study of Zhao and the team [1], could be due to species variation affecting the degree of hepcidin effect on bone metabolism. Hepcidin exerts its synergistic effect of upregulating the target genes of OC such as, CTK, MMP9 and NFATc1 which are needed in bone resorption activity of OC while inhibiting FPN1 which is well known iron exporter from inside to outside the cell. By confirming our previous research findings PPS deters osteoclastogenesis in a dose dependent manner at the range of concentration 1–20 μg/mL. The outcome of studies suggests that the inhibitory action of PPS over OC differentiation and function could be applied in treatment of pathological bone disorders such as osteoporosis or inflammatory arthritis where OC plays a vital role.

Current study data distinguished the novel capability of PPS by allowing it to interact with hepcidin at the process of canine bone-derived OC differentiation. Former research data demonstrated that the ability of hepcidin on enhancing differentiation and intracellular iron in OC with the use of mouse monocyte/macrophage cell line (RAW264.7) that can differentiate into multinucleated cells with an osteoclastic phenotype under the induction of RANKL [1, 38]. The results of previous study suggested that PPS at concentrations of 1 and 5 μg/mL suppressed M-CSF and RANKL-induced bone resorption activity and formation of actin-rings in matured OC (S1 Fig). The resorption lacunae, pit formation and actin ring formation are essential for OC bone resorption and is the most obvious character of mature OC during osteoclastogenesis [26, 39, 40]. With consisting of the former data on effect of PPS, our current findings further imply that hepcidin-facilitated bone resorption of OC can be condensed with the presence of 5, 10 μg/mL concentrated PPS in significant manner.

In our study, PPS at higher concentrations significantly suppressed the cathepsin K genomic expression in hepcidin-treated OC while reducing the MMP9 and NFATc1 expression in dose dependent manner. Once the Mitogen-activated protein kinase signaling cascade is activated, NFATc1 is activated as a master transcription factor for OC differentiation [30, 41]. Further, NFATc1 plays a dynamic role in upregulating expressions of genes required for OC maturation, such as cathepsin K and MMP-9 which are requisite for the bone resorption processes mediated by mature OC [42]. We further speculated that the existence of FPN1 at the membrane premature OC cells, and the expression of FPN1 was markedly downregulated by hepcidin in a concentration dependent manner, markedly at the 800 nmol/L. Intriguingly, PPS is able to inhibit the hepcidin-induced internalization and degradation of FPN1 molecule and that was visualized in gene expression level and by immunocytochemistry. With the increment of hepcidin, it binds to FPN1 molecules and provokes their internalization and degradation, and iron release is decreased progressively [3]. With that inhibition of degradation of FPN1 molecule in OC by PPS, facilitate the iron releasing into the outside of OC cells.

Some of the limitations of our study are the not analyzing the protein level FPN expression. We believe that current presentation of data would be a good platform to initiate protein level work together with advanced protein analysis for each factors, focused on this study other than the current finding of specific gene expression level for OC to understanding how the PPS affect those cells in transcriptional level. To the author's knowledge, no similar studies have previously been reported to determine a priori what the crucial pathways to address. Current study took place to explain the fundamentals of the PPS effect in cell culture level using changes of differentiation and functional changes associated with osteoclasts. Further, undifferentiated bone marrow mononuclear cells (BMMs or osteoclast precursor cells) do not show reduced FPN levels after hepcidin treatment could be the true factor hence BMM cells might

have comparatively less hepcidin-induced FPN1 internalization and degradation compared to mature osteoclasts, which needs additional step forward to examine the variability of cellular response to hepcidin and understand the variability of metabolism.

Previous studies have indicated that OC development comprises high iron requirements [43]. With that phenomenal statement, it further supplemented the inhibitory effect of PPS on osteoclastogenesis and their functional capability even at the presence of hepcidin. Outcome of this study confirms that PPS regulates hepcidin 1-facilitated formation and function of OC derived from canine bone marrow thereby inhibiting iron accumulation in the cells. However, further investigations would be required to clarify the mechanism of action of PPS on hepcidin inhibition. This *in vitro* study will pave the other clinical aspect of PPS to explore its additional therapeutic application against hepcidin compound which plays a vital role in the anemia of inflammation observed in many RA patients.

## Supporting information

**S1 Fig. Osteoclasts actin ring formation (Rhodamin Phalloidin) was monitored after treatment of PPS with different concentartions.** Osteoclasts derived from canine bone marrow were treated with various concentrations (0, 5, 10, 20 μg/mL) of PPS followed by M-CSF (20 ng/mL) and RANKL (50 ng/mL) for 7 days stained with phalloidin, which detects filamentous actin. An actin ring is a characteristic actin structure that is essential for bone resorption by osteoclasts. Scale bar- 100 μm. PPS showed inhibitory effect of formation of number of acting ring and the differention of osteoclast in a concentartion dependent manner.
(TIFF)

**S1 Table. Primer sequences used to polymerize the osteoclast specific genes.**
(DOCX)

## Author Contributions

**Conceptualization:** Suranji Wijekoon, Takafumi Sunaga, Yanlin Wang, Carol Mwale, Sangho Kim, Masahiro Okumura.

**Data curation:** Suranji Wijekoon, Takafumi Sunaga, Masahiro Okumura.

**Formal analysis:** Suranji Wijekoon.

**Funding acquisition:** Suranji Wijekoon, Takafumi Sunaga, Sangho Kim, Masahiro Okumura.

**Investigation:** Suranji Wijekoon, Yanlin Wang, Carol Mwale.

**Methodology:** Suranji Wijekoon, Takafumi Sunaga, Masahiro Okumura.

**Project administration:** Suranji Wijekoon, Takafumi Sunaga, Sangho Kim, Masahiro Okumura.

**Resources:** Suranji Wijekoon, Takafumi Sunaga, Sangho Kim, Masahiro Okumura.

**Software:** Suranji Wijekoon.

**Supervision:** Suranji Wijekoon, Masahiro Okumura.

**Validation:** Suranji Wijekoon, Takafumi Sunaga, Yanlin Wang, Carol Mwale, Sangho Kim, Masahiro Okumura.

**Visualization:** Suranji Wijekoon, Takafumi Sunaga, Yanlin Wang, Carol Mwale, Sangho Kim, Masahiro Okumura.

**Writing – original draft:** Suranji Wijekoon.

**Writing – review & editing:** Suranji Wijekoon, Masahiro Okumura.

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
