## [Decision Letter · Decision Letter 0]

1 Dec 2021

PONE-D-21-31424Pentosan polysulfate regulates hepcidin 1-facilitated formation and function of osteoclast derived from canine bone marrowPLOS ONE

Dear Dr. Wijekoon,

Thank you for submitting your manuscript to PLOS ONE. After careful consideration, we feel that it has merit but does not fully meet PLOS ONE’s publication criteria as it currently stands. Therefore, we invite you to submit a revised version of the manuscript that addresses the points raised during the review process.Please ensure that your decision is justified on PLOS ONE’s publication criteria and not, for example, on novelty or perceived impact.

We look forward to receiving your revised manuscript.

Kind regards,

Dominique Heymann, Ph.D.

Academic Editor

PLOS ONE

Journal Requirements:

2. Thank you for submitting the above manuscript to PLOS ONE. During our internal evaluation of the manuscript, we found significant text overlap between your submission and the following previously published works, some of which you are an author.

https://www.frontiersin.org/articles/10.3389/fphar.2016.00160/full

https://bmcvetres.biomedcentral.com/articles/10.1186/s12917-018-1466-4

Please revise the manuscript to rephrase the duplicated text, cite your sources, and provide details as to how the current manuscript advances on previous work. Please note that further consideration is dependent on the submission of a manuscript that addresses these concerns about the overlap in text with published work.

We will carefully review your manuscript upon resubmission, so please ensure that your revision is thorough

Reviewers' comments:

Reviewer's Responses to Questions

**Comments to the Author**

1. Is the manuscript technically sound, and do the data support the conclusions?

Reviewer #1: Yes

Reviewer #2: Yes

2. Has the statistical analysis been performed appropriately and rigorously? 

Reviewer #1: No

Reviewer #2: Yes

3. Have the authors made all data underlying the findings in their manuscript fully available?

Reviewer #1: Yes

Reviewer #2: Yes

4. Is the manuscript presented in an intelligible fashion and written in standard English?

Reviewer #1: Yes

Reviewer #2: Yes

5. Review Comments to the Author

Reviewer #1: In this manuscript, the authors demonstrate, in vitro, that pentosan polysulfate impairs hepcidin 1-induced osteoclast formation and function. Data is nicely presented and organized.

Flaws that should be addressed by the authors include:

- the animal model choice is not clear. What is the relevance of the canine model for that study? Could not the authors have used PBMC for such purpose? Or rodent bone marrow cells?

- no statistical analysis was reported, yet the authors used asterisks in graphs. Would that indicate statistical significance diferences? Which tests were applied? Gene expression show small error bars, but no asterisks. Please indicate methods used.

- manuscript present several grammar and typos errors. Please revise.

Reviewer #2: Hepcidin, a peptide hormone released mainly by liver hepatocytes, acts as a key regulator of systematic iron homeostasis. This function is achieved by hepcidin binding to ferroportin (FPN), the iron exporter that high expressed in macrophages and intestinal cells. FPN is internalized and degraded upon binding to the hepcidin, leading to a decrease in the export of intracellular iron from macrophages and intestinal cells. Hepcidin treatment thus increases intracellular iron levels and promotes proliferation and osteoclast differentiation of RAW264.7 cells (PMID: 25059214; PMID: 34108442). In the current manuscript, Wijekoon et al., found that pentosane polysulfate (PPS), a heparin analogue, could inhibit osteoclast differentiation during hepcidin treatment. Further studies showed reduced mRNA levels in Cathepsin K, MMP9, and NFATC1, but an increase in FPN1. In contrast, immunofluorescence staining showed decreased FPN1 levels and increased iron levels in osteoclasts after hepcidin treatment; and this hepcidin's effects could be reversed by PPS treatment. These results lead to the conclusion that PPS might be beneficial for treatment of OA and osteoporosis via its inhibitory effect on hepcidin. This study is of interest to the field. However, several concerns remain to be resolved, which are listed below.

1. Hepcidin has been reported to promote the proliferation and differentiation of osteoclast (OC) precursor cells (PMID: 25059214; PMID: 34108442). In addition to inhibiting OC differentiation, does PPS treatment affect the proliferation of OC precursor cells?

2. Some experiments lack necessary control groups. PPS treatment could inhibit OC differentiation without hepcidin treatment (in Fig 2). What is the underlying mechanism? It is better to include PPS treatments at different doses (e.g., 1, 5, 10 μg/mL) without hepcidin in Fig 3 and Fig 4, so that the effect of PPS on OC differentiation and gene expression with or without hepcidin can be compared.

3. In Fig 4D, are there any significant differences in the transcription levels of MMP9, NFATc1 and FPN1? It is better to use Q-PCR here, which is more accurate than that of the reverse transcription PCR. In addition, hepcidin is known to regulate the stability of FPN1 protein. How is the transcription level of FPN1 increased after PPS treatment? Does PPS regulate FPN1 expression via a hepcidin-independent manner?

4. The published paper and the current manuscript have shown that hepcidin promotes the differentiation of OC precursor cells. In Fig 5, it is better to examine the FPN and iron levels in OC precursor cells. In Fig 5, what are the small size cells surrounding the OCs? If they are the undifferentiated bone marrow mononuclear cell (BMMs), why they did not show reduced FPN levels after hepcidin treatments? The FPN and iron levels were also not changed in these cells after PPS treatments (Fig 5D-G).

5. Western blot is a more accurate method to detect the expression level of FPN protein (or MMP9, NFATc1 and so on).

6. If PPS regulates OC differentiation via changing FPN or iron levels, it would be of interest to add iron mimic such as FAC to see whether it can diminish the effects by PPS.

7. The OCs numbers in Fig1 and Fig3 do not appear to have significant changes. It would be more convincing to show the entire well instead of the enlarged field of the view.

6. PLOS authors have the option to publish the peer review history of their article (what does this mean?). If published, this will include your full peer review and any attached files.

Reviewer #1: No

Reviewer #2: No

---

## [Author Response · Author response to Decision Letter 0]

19 Dec 2021

Dear Editor-in-Chief and Reviewers, 

Subject: Submission of revised paper 

Thank you for your email dated 1st December 2021 enclosing reviewer’s comments. We have carefully reviewed the comments and have revised the manuscript accordingly. Our responses are given in a point-by-point manner below. 

Thank you for your consideration of this manuscript. We hope the revised version is now suitable for publication and look forward to hearing from you in due time regarding our submission and to respond to any further questions and comments you may have. Thank you for your consideration of this manuscript. 

suranjisk@gmail.com

Sincerely,

Dr. Suranji Wijekoon (PhD., Mphill., BVSc., MSLCVS)

(Corresponding Author)

Response to reviewer 1:

Thank you for your review of our paper. We have answered each of your points below.

[In this manuscript, the authors demonstrate, in vitro, that pentosan polysulfate impairs hepcidin 1-induced osteoclast formation and function. Data is nicely presented and organized] 

Thank you for the valuable comment

[The animal model choice is not clear. What is the relevance of the canine model for that study? Could not the authors have used PBMC for such purpose? Or rodent bone marrow cells]

Thank you very much for the comment. 

Spontaneous inflammatory arthropathies might be a good model for human juvenile rheumatoid arthritis are available for further clinical perspectives on use of PPS. Enough amount for reliably repeatable use of bone marrow could be obtained from dogs for implication to therapeutic use of PPS to arthritis. It is known that the monocyte/macrophage lineage gives rise to osteoclast. Cells in the bone marrow are mainly precusors that upon maturation migrate out of the bone marrow and enter the circulation, hence it contains high amount of precursor cells. 

[No statistical analysis was reported, yet the authors used asterisks in graphs. Would that indicate statistical significance diferences? Which tests were applied? Gene expression show small error bars, but no asterisks. Please indicate methods used]

Thank you for the valuable comment. 

Data were analyzed using Statistical Package for the Social Sciences v.16 (SPSS inc., Chicago, IL, USA). Statistical significant of quantitative qPCR data, number of OC and resorption pits were determined by analysis of variance (ANOVA), comparing the mean values of the treatments. Where significant different observed, multiple comparison of group means was performed using Post Hoc Bonferroni. The results were considered significant at a 95% confidence level (p < .05). All quantitative results are presented as mean ± SE. Fig 4D indicates the gene expression levels of Quantitative qPCR data showed that PPS at 5-10 μg/mL significantly downregulate the gene expression (p<0.05) level of CTK while inhibiting expression level of MMP-9 and NFATc1 in concentration-dependent manner (figure 4D). However, the expression of FPN1 was upregulated toward the higher concentrations of PPS.

[manuscript present several grammar and typos errors. Please revise]

Thank you for the valuable comment. All the grammar and typo errors were addressed. 

Response to reviewer 2:

[This study is of interest to the field. However, several concerns remain to be resolved, which are listed below.]

Thank you for the valuable comment

1. [Hepcidin has been reported to promote the proliferation and differentiation of osteoclast (OC) precursor cells (PMID: 25059214; PMID: 34108442). In addition to inhibiting OC differentiation, does PPS treatment affect the proliferation of OC precursor cells?]

Thank you very much for your question. We will briefly give the background information of effect of PPS.

We have conducted several studies to understand the PPS activity on OC differentiation, proliferation, and function. All findings were already published in reputed journals. In certain studies, PPS at concentration of 5 μg/mL exerted an inhibitory effect on canine osteoclastogenesis through suppression of key transcription factors such as NFATc1, c-Fos while visualizing co-localization patterns. This information may partially support the suggestion that PPS may exert its inhibitory effect on OC by direct interaction with transcription factors, subsequently deterring the target genes like Cathepsin K and MMP-9 which are needed in bone resorption activity of OC. To further study the effects of PPS on osteoclastogenesis, we examined whether PPS affected RANKL-induced OC function by bone resorption assays and actin formation (Functional structure of OC). The results suggested that PPS at concentrations of 1 and 5 μg/mL suppressed RANKL-induced bone resorption activity and formation of actin-rings of matured OC. The stimulation of M-CSF and RANKL make mature OC result in resorption lacunae, pit formation and actin ring formation which is a prerequisite for OC bone resorption and is the most obvious character of mature OC during osteoclastogenesis. The outcome of this study suggests that the inhibitory action of PPS over OC differentiation and function could be applied in treatment of pathological bone disorders where OC play central role. Intracellular colocalization and interaction of PPS with c-Jun transcriptional factor were observed in this study by immunofluorescence assay emphasizing that the site of action of drug of interest. Binding of c-Fos to the NFATc1 promoter is important for its activation. Suppression of NFATc1 by PPS is the consequence of the downregulation of c-Fos, with the subsequent down-regulation of AP-1 activity and attenuation of OC–specific gene expression required for efficient OC differentiation and bone resorption. Some other finding related to efficacy of PPS over the osteoclast functional unit (actin ring) is added into supplementary files. With these all finding, we concluded that PPS could inhibit the proliferative ability of OC by affecting target genes and functional units. 

2. [Some experiments lack necessary control groups. PPS treatment could inhibit OC differentiation without hepcidin treatment (in Fig 2). What is the underlying mechanism? It is better to include PPS treatments at different doses (e.g., 1, 5, 10 μg/mL) without hepcidin in Fig 3 and Fig 4, so that the effect of PPS on OC differentiation and gene expression with or without hepcidin can be compared.]

Thank you very much for the comments and valuable suggestions. However, we have already demonstrated the effect of PPS on OC formation and functions. And those were already published. https://www.ncbi.nlm.nih.gov/pmc/articles/PMC5930774/. Briefly, we are explaining the underlying mechanism of OC inhibition by PPS. Intracellular colocalization and interaction of PPS with c-Jun transcriptional factor were observed in our previous study by immunofluorescence assay emphasizing that the site of action of drug of interest. Binding of c-Fos to the NFATc1 promoter is important for its activation. Suppression of NFATc1 by PPS is the consequence of the down-regulation of c-Fos, with the subsequent down-regulation of activator protein 1 transcription factor (AP-1) activity and attenuation of OC–specific gene expression required for efficient OC differentiation and bone resorption. Further extension of the study up to detailed work by evaluating specific binding affinity of PPS with specific protein at nuclear, sub nuclear domain or nuclear speckles in OC would be much awarded the PPS as therapeutic perspective.

To avoid the repetition of those findings, we displayed the effect of PPS in different doses (e.g., 1, 5, 10 μg/mL) without hepcidin in fig 2. We have sited previous experiments in this regard to avoid the confusion of readers. And compared the current results with previous findings comparing all most all the aspects. But thank you again for pointing this out. 

3. In Fig 4D, are there any significant differences in the transcription levels of MMP9, NFATc1 and FPN1? It is better to use Q-PCR here, which is more accurate than that of the reverse transcription PCR. In addition, hepcidin is known to regulate the stability of FPN1 protein. How is the transcription level of FPN1 increased after PPS treatment? Does PPS regulate FPN1 expression via a hepcidin-independent manner?

Thank you for the valuable comments and agreed with your statement. 

In figure legend that was mentioned as RT PCR, but it was correctly motioned in material and method. That was corrected and rephased. Thank you. All quantitative results are presented as mean ± SE. Fig 4D indicates the gene expression levels of Quantitative qPCR data showed that PPS at 5-10 μg/mL significantly downregulate the gene expression (p<0.05) level of CTK while inhibiting expression level of MMP-9 and NFATc1 in concentration-dependent manner (figure 4D). However, the expression of FPN1 was upregulated toward the higher concentrations of PPS.

4. [The published paper and the current manuscript have shown that hepcidin promotes the differentiation of OC precursor cells. In Fig 5, it is better to examine the FPN and iron levels in OC precursor cells. In Fig 5, what are the small size cells surrounding the OCs? If they are the undifferentiated bone marrow mononuclear cell (BMMs), why they did not show reduced FPN levels after hepcidin treatments? The FPN and iron levels were also not changed in these cells after PPS treatments (Fig 5D-G).]

Thank you for the comment and suggestions for advancement of future experiments. 

However, in this study we basically focused on how the PPS act on Hepcidin treated OC at the differentiating and matured stages where they are actively involved in bone resorption in osteoporosis, osteoarthritis, and rheumatoid arthritis. Small cells surrounding the OC are undifferentiated mononuclear cells which remained in the culture slides even after the 7 days. If we compare those mononuclear cells to mature OC, we can’t see very clear FPN changes. But if we carefully see those small cells in the Fig 5.CD, fluorescence intensities are slightly increased combined with PPS treatment. Interesting point in this juncture is selectivity of treatment efficacy of PPS which profusely shows the therapeutic application against OC among other cells. Those mature OC occupy a large area with several nuclei and cells are massive to visualize the changes compared to tiny cells. However, as you suggested it`s a very valuable point to look for the FPN and iron changes quantitatively within OC precursor cells. 

5.[Western blot is a more accurate method to detect the expression level of FPN protein (or MMP9, NFATc1 and so on).]

Thank you for the suggestion and agreed with it. We definitely include those additional tests to our future experiments to continue this study to further clarify the mode of action of PPS with regards to the iron metabolism in high demanding cells. 

 6.[If PPS regulates OC differentiation via changing FPN or iron levels, it would be of interest to add iron mimic such as FAC to see whether it can diminish the effects by PPS.]

Thank you for the comment. 

It very true and we were initially planned to add iron mimic to see how PPS works on it. However, we have already identified and confirmed the efficacy of PPS over the osteoclast genesis and function of OC. Here we more focused on the hepcidin facilitated OC formation and how PPS react over it. This current data very clearly emphasized the ability of PPS on regulating the hepcidin induced OC formation and function. With those fundamentals we will move further to understand the detailed molecular aspects of PPS effect on iron metabolism. 

7. [The OCs numbers in Fig1 and Fig3 do not appear to have significant changes. It would be more convincing to show the entire well instead of the enlarged field of the view.]

Thank you for the comment and agreed well with it. 

We have changed Fig 1 and Fig 3 according to the suggestion. Most profusely Fig 3 was changed adding entire well. 

Thank you very much for very constructive and supportive feedback to improve the current layout and to advance the research into the next level in future studies.

---

## [Decision Letter · Decision Letter 1]

21 Jan 2022

PONE-D-21-31424R1

Pentosan polysulfate regulates hepcidin 1-facilitated formation and function of osteoclast derived from canine bone marrow

PLOS ONE

Dear Dr. Wikekoon,

Thank you for submitting your manuscript to PLOS ONE. After careful consideration, we have decided that your manuscript does not meet our criteria for publication and must therefore be rejected.

Specifically:

Several key concerns are not fully addressed. As specifiied by the reviewer 2, the revised manuscript only shows ferriportin (FPN)(a receptor of hepcidin) expression at transcriptional level, but not at protein level (Fig 4). The undifferentiated bone marrow mononuclear cells (BMMs or osteoclast precursor cells) do not show reduced FPN levels after hepcidin treatment (Fig 5).

More important and that is a big issue, ame images appear duplicated and used in different figures. The “PPS 10 μg/mL” group in Fig 2 is similar to “Hepcidin 800 nmol/L + PPS 1 μg/mL” group in Fig 3, and the “PPS 20 μg/mL” group in Fig 2 is similar to “Hepcidin 800 nmol/L + PPS 5 μg/mL” group in Fig 3. It is then extremely complicated to determine which are the true data.

 I am sorry that we cannot be more positive on this occasion, but hope that you appreciate the reasons for this decision.Yours sincerely,

Dominique Heymann, Ph.D.

Academic Editor

PLOS ONE

Reviewers' comments:

Reviewer's Responses to Questions

**Comments to the Author**

1. If the authors have adequately addressed your comments raised in a previous round of review and you feel that this manuscript is now acceptable for publication, you may indicate that here to bypass the “Comments to the Author” section, enter your conflict of interest statement in the “Confidential to Editor” section, and submit your "Accept" recommendation.

Reviewer #1: All comments have been addressed

Reviewer #2: (No Response)

2. Is the manuscript technically sound, and do the data support the conclusions?

Reviewer #1: Yes

Reviewer #2: Partly

3. Has the statistical analysis been performed appropriately and rigorously? 

Reviewer #1: Yes

Reviewer #2: I Don't Know

4. Have the authors made all data underlying the findings in their manuscript fully available?

Reviewer #1: Yes

Reviewer #2: Yes

5. Is the manuscript presented in an intelligible fashion and written in standard English?

Reviewer #1: Yes

Reviewer #2: Yes

6. Review Comments to the Author

Reviewer #1: The authors have provided answers to all questions and made the changes in the manuscript accordingly.

Reviewer #2: The revised manuscript has addressed some, but not all of the concerns raised previously. Several key concerns are not fully addressed/ For examples, the revised manuscript only showed ferriportin (FPN)(a receptor of hepcidin) expression at transcriptional level, but not protein level (Fig 4). The undifferentiated bone marrow mononuclear cells (BMMs or osteoclast precursor cells) did not shown reduced FPN levels after hepcidin treatment (Fig 5). Additionally, same images appeared to be used in different figures. The “PPS 10 μg/mL” group in Fig 2 appeared to be same as the “Hepcidin 800 nmol/L + PPS 1 μg/mL” group in Fig 3, and the “PPS 20 μg/mL” group in Fig 2 appeared to be same data as the “Hepcidin 800 nmol/L + PPS 5 μg/mL” group in Fig 3. These issues may result in an incorrect quantification and conclusion.

7. PLOS authors have the option to publish the peer review history of their article (what does this mean?). If published, this will include your full peer review and any attached files.

Reviewer #1: No

Reviewer #2: No

- - - - -

---

## [Author Response · Author response to Decision Letter 1]

17 Feb 2022

H.M. Suranji Wijekoon

Graduate School of Veterinary Medicine

Department of Veterinary Clinical Sciences

Laboratory of Veterinary Surgery

Kita 18, Nishi 9, Kita-Ku, (060-0818), Sapporo

Hokkaido, Japan

31.01.2021

Dear Editor-in-Chief,

Subject: Submission of revised paper 

Thank you for your email dated 1st December 2021 enclosing reviewer’s comment and 26th dated information requested after considering appeal. We have carefully reviewed the comments and have revised the manuscript accordingly. Our responses are given in a point-by-point manner below for the concerns raised by the reviewers and Academic Editor and details on the revisions carried out on the manuscript since its original submission.

Data Availability: 

All relevant data are within the paper and its supporting information files.

Enclosed is a manuscript to be considered for publication in PLOSONE. The research reported in this manuscript has been ethically clearance with obtaining approval from Graduate School of Veterinary Medicine, Hokkaido University (approval No 21-0022) and no financial or personal conflicts of interest.

We confirm that this work is original and has not been published elsewhere nor is it currently under consideration for publication elsewhere.

Thank you for your consideration of this manuscript. We hope the revised version is now suitable for publication and look forward to hearing from you in due time regarding our submission and to respond to any further questions and comments you may have. Thank you for your consideration of this manuscript. 

suranjisk@gmail.com

Sincerely,

Dr. Suranji Wijekoon (PhD., Mphill., BVSc., MSLCVS)

(Corresponding Author)

Response to reviewer 1:

Thank you for your review of our paper. We have answered each of your points below.

[In this manuscript, the authors demonstrate, in vitro, that pentosan polysulfate impairs hepcidin 1-induced osteoclast formation and function. Data is nicely presented and organized] 

Thank you for the valuable comment

[The animal model choice is not clear. What is the relevance of the canine model for that study? Could not the authors have used PBMC for such purpose? Or rodent bone marrow cells]

Thank you very much for the comment. 

Spontaneous inflammatory arthropathies might be a good model for human juvenile rheumatoid arthritis are available for further clinical perspectives on use of PPS. Enough amount for reliably repeatable use of bone marrow could be obtained from dogs for implication to therapeutic use of PPS to arthritis. It is known that the monocyte/macrophage lineage gives rise to osteoclast. Cells in the bone marrow are mainly precusors that upon maturation migrate out of the bone marrow and enter the circulation, hence it contains high amount of precursor cells. 

[No statistical analysis was reported, yet the authors used asterisks in graphs. Would that indicate statistical significance diferences? Which tests were applied? Gene expression show small error bars, but no asterisks. Please indicate methods used]

Thank you for the valuable comment. 

Data were analyzed using Statistical Package for the Social Sciences v.16 (SPSS inc., Chicago, IL, USA). Statistical significant of quantitative qPCR data, number of OC and resorption pits were determined by analysis of variance (ANOVA), comparing the mean values of the treatments. Where significant different observed, multiple comparison of group means was performed using Post Hoc Bonferroni. The results were considered significant at a 95% confidence level (p < .05). All quantitative results are presented as mean ± SE. Fig 4D indicates the gene expression levels of Quantitative qPCR data showed that PPS at 5-10 μg/mL significantly downregulate the gene expression (p<0.05) level of CTK while inhibiting expression level of MMP-9 and NFATc1 in concentration-dependent manner (figure 4D). However, the expression of FPN1 was upregulated toward the higher concentrations of PPS.

[manuscript present several grammar and typos errors. Please revise]

Thank you for the valuable comment. All the grammar and typo errors were addressed. 

Response to reviewer 2:

[This study is of interest to the field. However, several concerns remain to be resolved, which are listed below.]

Thank you for the valuable comment. We believe that we have addressed almost all the concerns with very detailed explanation, rewrite the certain part of manuscript agreeing all most all the valuable comments and further clarified the few concerns by elucidating future study expansion based on current findings.

1. [Hepcidin has been reported to promote the proliferation and differentiation of osteoclast (OC) precursor cells (PMID: 25059214; PMID: 34108442). In addition to inhibiting OC differentiation, does PPS treatment affect the proliferation of OC precursor cells?]

Thank you very much for your question. We will briefly give the background information of effect of PPS.

We have conducted several studies to understand the PPS activity on OC differentiation, proliferation, and function. All findings were already published in reputed journals. In certain studies, PPS at concentration of 5 μg/mL exerted an inhibitory effect on canine osteoclastogenesis through suppression of key transcription factors such as NFATc1, c-Fos while visualizing co-localization patterns. This information may partially support the suggestion that PPS may exert its inhibitory effect on OC by direct interaction with transcription factors, subsequently deterring the target genes like Cathepsin K and MMP-9 which are needed in bone resorption activity of OC. To further study the effects of PPS on osteoclastogenesis, we examined whether PPS affected RANKL-induced OC function by bone resorption assays and actin formation (Functional structure of OC). The results suggested that PPS at concentrations of 1 and 5 μg/mL suppressed RANKL-induced bone resorption activity and formation of actin-rings of matured OC. The stimulation of M-CSF and RANKL make mature OC result in resorption lacunae, pit formation and actin ring formation which is a prerequisite for OC bone resorption and is the most obvious character of mature OC during osteoclastogenesis. The outcome of this study suggests that the inhibitory action of PPS over OC differentiation and function could be applied in treatment of pathological bone disorders where OC play central role. Intracellular colocalization and interaction of PPS with c-Jun transcriptional factor were observed in this study by immunofluorescence assay emphasizing that the site of action of drug of interest. Binding of c-Fos to the NFATc1 promoter is important for its activation. Suppression of NFATc1 by PPS is the consequence of the downregulation of c-Fos, with the subsequent down-regulation of AP-1 activity and attenuation of OC–specific gene expression required for efficient OC differentiation and bone resorption. Some other finding related to efficacy of PPS over the osteoclast functional unit (actin ring) is added into supplementary files. With these all finding, we concluded that PPS could inhibit the proliferative ability of OC by affecting target genes and functional units. 

2. [Some experiments lack necessary control groups. PPS treatment could inhibit OC differentiation without hepcidin treatment (in Fig 2). What is the underlying mechanism? It is better to include PPS treatments at different doses (e.g., 1, 5, 10 μg/mL) without hepcidin in Fig 3 and Fig 4, so that the effect of PPS on OC differentiation and gene expression with or without hepcidin can be compared.]

Thank you very much for the comments and valuable suggestions. However, we have already demonstrated the effect of PPS on OC formation and functions. And those were already published. https://www.ncbi.nlm.nih.gov/pmc/articles/PMC5930774/. Briefly, we are explaining the underlying mechanism of OC inhibition by PPS. Intracellular colocalization and interaction of PPS with c-Jun transcriptional factor were observed in our previous study by immunofluorescence assay emphasizing that the site of action of drug of interest. Binding of c-Fos to the NFATc1 promoter is important for its activation. Suppression of NFATc1 by PPS is the consequence of the down-regulation of c-Fos, with the subsequent down-regulation of activator protein 1 transcription factor (AP-1) activity and attenuation of OC–specific gene expression required for efficient OC differentiation and bone resorption. Further extension of the study up to detailed work by evaluating specific binding affinity of PPS with specific protein at nuclear, sub nuclear domain or nuclear speckles in OC would be much awarded the PPS as therapeutic perspective.

To avoid the repetition of those findings, we displayed the effect of PPS in different doses (e.g., 1, 5, 10 μg/mL) without hepcidin in fig 2. We have sited previous experiments in this regard to avoid the confusion of readers. And compared the current results with previous findings comparing all most all the aspects. But thank you again for pointing this out. 

3. In Fig 4D, are there any significant differences in the transcription levels of MMP9, NFATc1 and FPN1? It is better to use Q-PCR here, which is more accurate than that of the reverse transcription PCR. In addition, hepcidin is known to regulate the stability of FPN1 protein. How is the transcription level of FPN1 increased after PPS treatment? Does PPS regulate FPN1 expression via a hepcidin-independent manner?

Thank you for the valuable comments and agreed with your statement. 

In figure legend that was mentioned as RT PCR, but it was correctly motioned in material and method. That was corrected and rephased. Thank you. All quantitative results are presented as mean ± SE. Fig 4D indicates the gene expression levels of Quantitative qPCR data showed that PPS at 5-10 μg/mL significantly downregulate the gene expression (p<0.05) level of CTK while inhibiting expression level of MMP-9 and NFATc1 in concentration-dependent manner (figure 4D). However, the expression of FPN1 was upregulated toward the higher concentrations of PPS.

4. [The published paper and the current manuscript have shown that hepcidin promotes the differentiation of OC precursor cells. In Fig 5, it is better to examine the FPN and iron levels in OC precursor cells. In Fig 5, what are the small size cells surrounding the OCs? If they are the undifferentiated bone marrow mononuclear cell (BMMs), why they did not show reduced FPN levels after hepcidin treatments? The FPN and iron levels were also not changed in these cells after PPS treatments (Fig 5D-G).]

Thank you for the comment and suggestions for advancement of future experiments. 

However, in this study we basically focused on how the PPS act on Hepcidin treated OC at the differentiating and matured stages where they are actively involved in bone resorption in osteoporosis, osteoarthritis, and rheumatoid arthritis. Small cells surrounding the OC are undifferentiated mononuclear cells which remained in the culture slides even after the 7 days. If we compare those mononuclear cells to mature OC, we can’t see very clear FPN changes. But if we carefully see those small cells in the Fig 5.CD, fluorescence intensities are slightly increased combined with PPS treatment. Interesting point in this juncture is selectivity of treatment efficacy of PPS which profusely shows the therapeutic application against OC among other cells. Those mature OC occupy a large area with several nuclei and cells are massive to visualize the changes compared to tiny cells. However, as you suggested it`s a very valuable point to look for the FPN and iron changes quantitatively within OC precursor cells. On the other hand undifferentiated bone marrow mononuclear cells (BMMs or osteoclast precursor cells) do not show reduced FPN levels after hepcidin treatment could be the true factor hence BMM cells might have comparatively less hepcidin-induced FPN1 internalization and degradation compared to mature osteoclasts. We also don’t know. We believe that it could be due to different level of metabolic capacity in the cells. And those undifferentiated cells need different time frame to react on those compounds. Osteoclasts are highly active cell types which carries higher grade metabolic activity rate compared to other cells. Those all novel findings will be addressed again with future progressive study and will apply protein work on top of that. 

5.[Western blot is a more accurate method to detect the expression level of FPN protein (or MMP9, NFATc1 and so on).]

Thank you for the suggestion and agreed with it. We definitely include those additional tests to our future experiments to continue this study to further clarify the mode of action of PPS with regards to the iron metabolism in high demanding cells. 

 6.[If PPS regulates OC differentiation via changing FPN or iron levels, it would be of interest to add iron mimic such as FAC to see whether it can diminish the effects by PPS.]

Thank you for the comment. 

It very true and we were initially planned to add iron mimic to see how PPS works on it. However, we have already identified and confirmed the efficacy of PPS over the osteoclast genesis and function of OC. Here we more focused on the hepcidin facilitated OC formation and how PPS react over it. This current data very clearly emphasized the ability of PPS on regulating the hepcidin induced OC formation and function. With those fundamentals we will move further to understand the detailed molecular aspects of PPS effect on iron metabolism. 

7. [The OCs numbers in Fig1 and Fig3 do not appear to have significant changes. It would be more convincing to show the entire well instead of the enlarged field of the view.]

Thank you for the comment and agreed well with it. 

We have changed the figure. 

Thank you very much for very constructive and supportive feedback to improve the current layout and to advance the research into the next level in future studies.

Appeal

Dear Editor, 

Receive the decision. Thank you very much. 

However I just want to clarify certain points regarding the facts you have considered to reject the following manuscript. Please kindly consider the explanation. Thank you for understanding. 

After reading the clarification we believe you will reconsider the decision OR you will just keep this in a side; however I respect the final decision whatever you will take. 

Please ask for further clarification if needed. Thank you for considering our paper. 

PONE-D-21-31424R1

Pentosan polysulfate regulates hepcidin 1-facilitated formation and function of osteoclast derived from canine bone marrow

PLOS ONE

Reviewer #1: All comments have been addressed

Reviewer #2: Raised few concerns. We have carefully reviewed the comments again and we identified one mistake in figure, happened in second submission when we responding the first revision. And other concern of Reviewer 2 was addressed, and we have responded in submission of revision. However, reviewer# 2 has mentioned that we haven’t addressed two points but accepted intelligible fashion of writing and written in standard English. 

Our responses are given in a point-by-point manner below for the two concerns raised by reviewer after we submitted the revision addressing totally 12 major issues in well manner. 

Specifically:

1- As specifiied by the reviewer 2, the revised manuscript only shows ferriportin (FPN)(a receptor of hepcidin) expression at transcriptional level, but not at protein level (Fig 4). The undifferentiated bone marrow mononuclear cells (BMMs or osteoclast precursor cells) do not show reduced FPN levels after hepcidin treatment (Fig 5).

Clarification; 

We have addressed all the key concerns of Reviewer 2 and however we respect this statement. In that response we have mentioned very clearly that ferriportin (FPN)(a receptor of hepcidin) expression in protein level was not analyzed in this study; however we are planning to add that technique together in the future work. Because the current study took place to explain the fundamentals of the PPS effect in cell culture level using changes of differentiation and functional changes associated with osteoclasts. Specific gene expression level for Osteoclasts were examined to understanding how the PPS affect those cells in transcriptional level. We have understood that the importance of protein level study but that will be included in our next step of the research for deeper understanding of the molecular aspects. As this is a very initial finding of the field of veterinary aspect additional steps will be taken progressively. We believe that current presentation of data would be a good platform to initiate protein level work as we clearly mentioned and submitted on previous submission of revision 1 `Response to reviewer` document. 

Further, undifferentiated bone marrow mononuclear cells (BMMs or osteoclast precursor cells) do not show reduced FPN levels after hepcidin treatment could be the true factor hence BMM cells might have comparatively less hepcidin-induced FPN1 internalization and degradation compared to mature osteoclasts. We also don’t know. We believe that it could be due to different level of metabolic capacity in the cells. And those undifferentiated cells need different time frame to react on those compounds. Osteoclasts are highly active cell types which carries higher grade metabolic activity rate compared to other cells. Those all novel findings will be addressed again with future progressive study and will apply protein work on top of that. We have given such a very clear explanation to Reviewer 2 and we wondered how it ended with the same comment. All other concerns of reviewer 2 were well addressed according to his statements.

2- More important and that is a big issue, ame images appear duplicated and used in different figures. The “PPS 10 μg/mL” group in Fig 2 is similar to “Hepcidin 800 nmol/L + PPS 1 μg/mL” group in Fig 3, and the “PPS 20 μg/mL” group in Fig 2 is similar to “Hepcidin 800 nmol/L + PPS 5 μg/mL” group in Fig 3. It is then extremely complicated to determine which is the true data. 

Clarification;

If you carefully reobserve our first submission before revision you will understand that the image issue is a mistake that happened while preparing images again for revision 1. Because that duplication was not there at the first submission and we are not in the level of making such a way unless there is a mistake. Hereby I again request to RECONSIDER that matter and hope you will understand the issue. Kindly recheck the first submission docs which you used to send to reviewer for their comments. Images were duplicated while revision.

Thank you very much for very constructive and supportive feedback to improve the current layout and to advance the research into the next level in future studies.

---

## [Decision Letter · Decision Letter 2]

7 Mar 2022

Pentosan polysulfate regulates hepcidin 1-facilitated formation and function of osteoclast derived from canine bone marrow

PONE-D-21-31424R2

Dear Dr. Dr Wijekoon,

We’re pleased to inform you that your manuscript has been judged scientifically suitable for publication and will be formally accepted for publication once it meets all outstanding technical requirements.

Kind regards,

Dominique Heymann, Ph.D.

Academic Editor

PLOS ONE

Additional Editor Comments (optional):

Reviewers' comments:

Reviewer's Responses to Questions

**Comments to the Author**

1. If the authors have adequately addressed your comments raised in a previous round of review and you feel that this manuscript is now acceptable for publication, you may indicate that here to bypass the “Comments to the Author” section, enter your conflict of interest statement in the “Confidential to Editor” section, and submit your "Accept" recommendation.

Reviewer #1: All comments have been addressed

2. Is the manuscript technically sound, and do the data support the conclusions?

Reviewer #1: Yes

3. Has the statistical analysis been performed appropriately and rigorously? 

Reviewer #1: Yes

4. Have the authors made all data underlying the findings in their manuscript fully available?

Reviewer #1: Yes

5. Is the manuscript presented in an intelligible fashion and written in standard English?

Reviewer #1: Yes

6. Review Comments to the Author

Reviewer #1: The authors have provided answers to all questions and made the changes in the manuscript accordingly.

7. PLOS authors have the option to publish the peer review history of their article (what does this mean?). If published, this will include your full peer review and any attached files.

Reviewer #1: No

---

## [Editor Report · Acceptance letter]

9 Mar 2022

PONE-D-21-31424R2 

Pentosan polysulfate regulates hepcidin 1-facilitated formation and function of osteoclast derived from canine bone marrow 

Dear Dr. Wijekoon:

I'm pleased to inform you that your manuscript has been deemed suitable for publication in PLOS ONE. Congratulations! Your manuscript is now with our production department. 

Kind regards, 

on behalf of

Pr. Dominique Heymann 

Academic Editor

PLOS ONE